# In Search of the Appropriate Anticoagulant-Associated Bleeding Risk Assessment Model for Cancer-Associated Thrombosis Patients

**DOI:** 10.3390/cancers14081937

**Published:** 2022-04-12

**Authors:** Géraldine Poénou, Emmanuel Tolédano, Hélène Helfer, Ludovic Plaisance, Florent Happe, Edouard Versini, Nevine Diab, Sadji Djennaoui, Isabelle Mahé

**Affiliations:** 1Médecine Interne, Hôpital Louis Mourier, Assistance Publique Hôpitaux de Paris, 92700 Colombes, France; emmanuel.toledano@aphp.fr (E.T.); helene.helfer@aphp.fr (H.H.); ludovic.plaisance@aphp.fr (L.P.); florent.happe@aphp.fr (F.H.); edouard.versini@aphp.fr (E.V.); nevine.diab@etu.u-paris.fr (N.D.); sadji.djennaoui@aphp.fr (S.D.); isabelle.mahe@aphp.fr (I.M.); 2Université de Paris Cité, 75006 Paris, France; 3Unité Inserm UMR-S1140 Innovation Thérapeutique en Hémostase, 75006 Paris, France; 4INNOVTE-FCRIN, CEDEX 2, 42055 Saint-Etienne, France

**Keywords:** cancer associated thrombosis, bleeding, risk assessment model

## Abstract

**Simple Summary:**

Patients with venous thromboembolism events in the context of cancer should receive anticoagulants as long as the cancer is active. Therefore, a tailor-made anticoagulation strategy should rely on an individualized assessment of the risks of recurrent venous thromboembolism and anticoagulant-associated bleeding. No existing risk assessment model for anticoagulant-associated bleeding risk has been validated for cancer-associated thrombosis. To obtain a better risk assessment model to assess anticoagulant-associated bleeding risk in cancer-associated thrombosis patients, we deemed it necessary to answer questions related to how and when to assess anticoagulant-associated bleeding risk as well as what factors to assess for which patients.

**Abstract:**

Patients with venous thromboembolism events (VTE) in the context of cancer should receive anticoagulants as long as the cancer is active. Therefore, a tailor-made anticoagulation strategy should rely on an individualized risk assessment model (RAM) of recurrent VTE and anticoagulant-associated bleeding. The aim of this review is to investigate the applicability of the currently available RAMs for anticoagulant-associated bleeding after VTE in the CAT population and to provide new insights on how we can succeed in developing a new anticoagulant-associated bleeding RAM for the current medical care of CAT patients. A systematic search for peer-reviewed publications was performed in PubMed. Studies, including systematic reviews, were eligible if they comprised patients with VTE and used a design for developing a prediction model, score, or other prognostic tools for anticoagulant-associated bleeding during anticoagulant treatment. Out of 15 RAMs, just the CAT-BLEED was developed for CAT patients and none of the presented RAMs developed for the VTE general population were externally validated in a population of CAT patients. The current review illustrates the limitations of the available RAMs for anticoagulant-associated bleeding in CAT patients. The development of a RAM for bleeding risk assessment in patients with CAT is warranted.

## 1. Introduction

Venous thromboembolism (VTE), which encompasses pulmonary embolism (PE) and deep vein thrombosis (DVT), is one of the most common complications in patients with cancer. PE is among the leading causes of death in cancer patients, and the occurrence of thromboembolic events is a negative prognostic factor beyond direct VTE-related mortality, underlining the strong relation between the hemostatic system and malignancy [1].

In cancer associated thrombosis (CAT), several treatment options are available. Until recently, the recommended treatment for CAT was low-molecular-weight heparin (LMWH), based on the CLOT trial where vitamin K antagonists (VKA) and LMWH (dalteparin) were compared. Within this study, a lower rate of recurrent VTE at 6 months (17% vs. 9%; HR: 0.48; 95% CI: 0.30 to 0.77) and a similar risk of bleeding events (6% vs. 4%, *p* = 0.27) were observed in patients treated with dalteparin compared with patients treated with VKA, respectively [2].

More recently, randomized, controlled trials were initiated to assess direct oral anticoagulants (DOAC) in CAT patient populations [3,4,5,6]. DOACs demonstrated non-inferiority for the efficacy endpoint (recurrent VTE) and variable rates of bleeding compared with dalteparin [3,4,5,6]. In this context, DOACs have been incorporated into international guidelines for the management and treatment of CAT as alternative to dalteparin after 6 months of treatment or when dalteparin is poorly tolerated [7,8,9].

Considering that recurrent VTE and major bleeding complications are associated with significant morbidity and a decrease in quality of life in patients with cancer, it is pivotal to weigh the risks and benefits to minimize the risk of these complications when deciding which anticoagulant should be prescribed and for how long [10,11]. An increased risk of recurrent VTE and bleeding complications among cancer patients complicates the management of VTE compared with that in patients without cancer [12]. On the one hand, there is up to a six times higher risk for anticoagulant-associated bleeding (13.3 vs. 2.1, respectively, per 100 patient-years) and on the other hand, there is up to a three times higher risk for VTE recurrence (27.1 vs. 9.0, respectively, per 100 patient-years) with treatment [13]. If, based on meta-analysis, we can all agree that the efficacy of DOACs and LMWH for preventing CAT is comparable, we can also notice the discrepancies for anticoagulant-associated bleeding risk that are specific to each anticoagulant and depend on the site of cancer [14]. Based on that observation, the anticoagulant-associated bleeding risk influences the choice of the anticoagulant.

Being able to determine the risk of anticoagulant-associated bleeding might impact the type, the duration, and/or the intensity of anticoagulation therapy. Because of the high bleeding risk, general guidelines for VTE treatment propose incorporating an assessment of bleeding risk in treatment decisions [15,16]. Several risk assessment models (RAM) for bleeding during anticoagulation have been developed in patients with VTE [17]. Cancer is a predictive factor in 10 out of 15 of these RAMs, which implies that these RAMs work in CAT patients [17].

Therefore, cancer patients with VTE constitute a unique population distinct from VTE patients without cancer. A better understanding of the associated risks and benefits of treatment is required to identify a personalized anticoagulation strategy, with a tailor-made anticoagulation option choice, anticoagulation posology, and duration of treatment that coincides with a significant improvement in clinical outcomes. Within the current systematic review, we aimed to summarize the different RAMs developed for VTE patients treated with anticoagulant drugs (except thrombolysis drugs) for bleeding risk and to investigate whether or not an appropriate RAM is feasible in CAT patients.

## 2. Materials and Methods

A systematic search for peer-reviewed publications published through 20 September 2021 was performed in PubMed. Specific systematic review questions were pre-specified (Table 1). The present study was registered in the “International Prospective Register of Systematic Review” (systematic review registration number: Prospero No. 42022297863) and was conducted according to PRISMA guidelines.

We combined search terms for (a) DVT or PE or VTE, (b) bleeding or anticoagulant-associated bleeding, and (c) model, prediction score, risk, prognosis, risk assessment, decision tree, prediction rule, and clinical decision rule. The search was restricted to titles and MeSH terms, articles in English, and no limitation in time period (Table 2). Reference lists of all eligible articles and systematic reviews were manually searched for additional studies. Search and selection were conducted by two researchers independently (GP, HH). Any disagreement was resolved by discussion with the co-authors.

Studies, including systematic reviews, were eligible if they comprised (a) patients with DVT and/or PE and (b) used a design for developing a prediction model, score, or other prognostic tools for anticoagulant-associated bleeding during anticoagulant treatment. Studies focusing on predicting the risk of anticoagulation-associated bleeding in a non-VTE based population were excluded. For this reason, we excluded seven models initially developed for bleeding risk assessment in atrial fibrillation (OBRI, mOBRI, Shireman, HEMORR2HAGES, HAS-BLED, ATRIA, and ORBIT) because even though they were externally validated in a VTE population, they performed poorly in the general VTE population [18,19,20,21,22]. In addition, studies focusing on VTE not defined as new-onset symptoms of DVT or PE with the final diagnosis confirmed by objective tests according to the current standard were excluded. Finally, the BACS RAM, developed to predict major bleeding in acute PE patients receiving systemic thrombolysis, was excluded because of its limited relevance to our review [23].

Data extraction was performed according to the CHARMS checklist by one researcher (GP) using a predesigned data extraction form and an assessment of risk of bias was performed with the PROBAST list [24] (Appendix A). Most studies on predicting bleeding used the definitions provided by the (International Society of Thrombosis and Hemostasis) ISTH or related definitions for major bleeding and clinically relevant non-major bleeding (CRNMB) [25,26] (Appendix B). No conflicts of interest are reported by the authors.

## 3. Results

### 3.1. Description of the Bleeding Risk Assessment Models in VTE Patients

Overall, 15 RAMs for anticoagulant-associated bleeding developed in VTE patients were identified after the systemic literature search (Figure 1, Table 3 and Table 4): 7 of the 15 prediction models were derived from the characteristics of populations selected for randomized, controlled trials. Several RAMs, especially those developed in populations from randomized, controlled trials, had heterogeneous populations containing patients with either unprovoked or provoked VTE. Only four RAMs were developed in patients receiving DOAC. Follow-up ranged from less than 1 month to 3 years, and sample size varied between 194 patients for the Nieuwenhuis RAM and 13957 patients for the RIETE RAM [27,28]. The proportion of CAT patients (except for the CAT-BLEED study) was non-majority and heterogeneous (2% for the VTE bleed vs. 18% in the Chopard RAM and the Skowrońska RAM) [29,30,31]. The VTE risk associated with the presence of a neoplasm was also diverse, from slightly protective in the Nieuwenhuis study (RR: 0.9, no CI reported) to a vastly prohemorrhagic factor in early post-VTE active cancer in the Martinez study (sHR: 7.92 (4.33–14.49)) [32]. Eleven models integrated cancer (history of cancer, active cancer, or metastatic cancer) as one of the risk factors for bleeding [16,27,29,32,33,34,35,36]. Recent attempts to validate RAMs in patients with VTE (irrespective of cancer) are summarized in Table 5, but it appears that the predictive value for bleeding events was zero to modest. The two most reliable RAMs for the general population are the VTE BLEED and the RIETE RAMs (Table 5). CAT patients are known to display an increased risk for anticoagulant-associated bleeding. Regarding the external validation of anticoagulant-associated bleeding in RAMs for CAT patients specifically, de Winter et al. used the randomized, controlled trial VTE HOKUSAI cancer population (97.8% of patients with active cancer) to test, among other RAMs, the VTE BLEED and the RIETE RAMs and found the same poor performance. In this context, the CAT-BLEED RAM was specifically designed. (Table 5). Indeed, the VTE HOKUSAI Cancer trial population allowed Winter et al. to more precisely calculate the added risk of the tumor site, particularly in genitourinary cancer (sHR: 2.48 (1.14–5.38)), the weight of the relation between gastrointestinal cancer and edoxaban treatment (sHR: 2.20 (1.07–4.53)), and the weight of the use of anticancer therapies associated with gastrointestinal toxicity (sHR: 1.74 (1.03–2.92)). Therefore, in the CAT-BLEED RAM, in addition to the presence of cancer as a risk factor, genitourinary cancer, the association of gastro intestinal cancer with edoxaban, and the administration of anticancer therapy with gastrointestinal toxicity were included as parameters considered to increase the risk of anticoagulant-associated bleeding. Unfortunately, since these observations stem from a randomized, controlled trial population, no conclusions can be drawn regarding CAT patients with active bleeding, significant kidney or liver disease, short life expectancy, or thrombocytopenia. At this time, the CAT-BLEED has not been externally validated. All the details regarding the development and the validation of the different RAMs are described in Table 4 and Table 5.

Included model derivation studies were generally at low risk of bias for ‘predictors’ and ‘outcome’. However, for the population, as these RAMs derived mainly from randomized trials, the percentage of CAT patients was often low and their specific risk was not analyzed. With regard to the “statistical analysis” domain, almost all the derivation studies included were judged to be at high risk of bias due to poor management of the parameters in continuous values and the low event rates. These commentaries are transposable to RAM validation studies.

### 3.2. Description of the Factors Predicting Anticoagulant-Associated Bleeding

Age and cancer appeared to be the two most important factors associated with anticoagulant-associated bleeding in 11 of the 15 RAMs (Table 3). History of bleeding or anemia occurred either as a risk factor for bleeding tendency or as an independent risk factor in 10 of the 15 RAMs and 8 of the 15 RAMs, respectively. Sex, thrombocytopenia, and cardiovascular disease (stroke/coronaropathy/peripheral arterial disease) were reported as predictive factors of anticoagulant-associated bleeding risk, although with weaker associations. Moreover, cardiovascular risk factors, such as uncontrolled hypertension or diabetes mellitus, irrespective of the use of antiplatelet therapy to prevent cardiovascular events, were also predictive factors of anticoagulant-associated bleeding risk. (Table 3)

Surprisingly, research of the literature showed that some factors have been demonstrated to increase the risk for both bleeding and thrombosis [18]. These factors include age, sex, BMI, PE as the index thrombotic event, history of cardiovascular disease (stroke, coronaropathy, peripheral arterial disease), cancer site, cancer stage, and chemotherapy, which emphasizes that these risks for bleeding and thrombosis should be considered at the time, not separately [18] (Table 6). This remarkable observation underlines that CAT patients need personalized decision-making depending on the cancer site, the cancer stage, and the anticancer therapy to minimize the risk of recurrent VTE or anticoagulant-associated bleeding.

Finally, to describe the most appropriate RAM to assess the anticoagulant-associated bleeding risk in CAT patients, we deemed it necessary to answer questions related to how and when to assess anticoagulant-associated bleeding risk as well as what factors to assess for which patients.

#### 3.2.1. Who Are the CAT Patients in Whom We Assess Anticoagulant-Associated Bleeding Risk?

The CAT patients for whom we assess the anticoagulant-associated bleeding risk have different characteristics compared with cancer-free patients with VTE. The Seiler and HOKUSAI RAMs are based on studies that excluded patients with a diagnosis of cancer at the time of inclusion. The rate of CAT patients in RAMs based on phase 3 trials on DOAC was below 6% (from 2.2% for VTE bleed RAM to 5.3% for EINSTEIN RAM). Also, it is important to note that in current medical care, CAT patients are different from VTE patients (even those with a diagnosed cancer) selected in phase 3 trials on DOAC used to build a RAM, such as the VTE bleed RAM, EINSTEIN RAM or HOKUSAI RAM. These patients were most likely initially at a lower risk of bleeding complications and with a better survival. In real practice, CAT patients can display the characteristics of a population that is systematically excluded from the randomized control trial, such as those on anticoagulants, those with creatinine clearance < 30 mL/min, clinically significant liver disease, ECOG 3–4, those with life expectancy < 3 months, or platelet counts < 50,000. Regarding the risk assessment for bleeding in CAT patients, RAMs do not systematically incorporate the risk of thrombocytopenia, the site and stage of the tumor or metastatic region, and drug–drug interactions. Thrombocytopenia is frequent in CAT patients due to the direct toxicity to the platelets induced by chemotherapy. The gastro-intestinal or genitourinary tracts, either as site of the tumor or as normal tissue, are well known risk sites for bleeding in cancer patients due to the effects of radiation, chemotherapy, or anticoagulants (mostly DOAC) [41]. Both the ISTH (International Society of Thrombosis and Hemostasis) and the ASCO (American Society of Clinical Oncology) recommend checking the risk of bleeding and drug–drug interactions when using edoxaban or rivaroxaban [7,8]. The chemotherapy–DOAC interaction in CAT patients might also be influenced by extreme weight, renal status, age, and digestive absorption modified by chemotherapy use (vomiting, nausea, diarrhea) [41]. The CAT-BLEED is the only RAM derived exclusively from a population of CAT patients. However, the CAT population used to derive the RAM was strictly selected for better survival criteria and therefore had a better prognosis than the average CAT patient. The appropriate RAM should be derived from and also be applicable to the population in which anticoagulation can be life threatening.

#### 3.2.2. How Do We Assess the Anticoagulant-Associated Bleeding Risk in CAT Patients?

It is important to define how the anticoagulant-associated bleeding risk in CAT patient is assessed and whether some differences should be taken into account rather than an overall assessment. Cancer is one of the most frequent parameters included in anticoagulant-associated bleeding RAMs. The RAMs categorize patients into low-, intermediate-, and high-risk categories for bleeding events. The presence of cancer puts patients in a higher risk class of anticoagulant-associated bleeding for 10 of the 14 RAMs developed in the general population. The VTE BLEED RAM is the only RAM that directly puts a patient with cancer in the higher risk category of bleeding. The four remaining RAMs that do not consider cancer as a predicting factor are the EINSTEIN, the HOKUSAI, the Nieuwenhuis and the Chopard RAMs.

In addition, the anticoagulant-associated bleeding risk depends on the anticoagulant used. In patients with VTE, it has been established that VKAs are responsible for a higher absolute risk of bleeding compared with DOACs (2.1% and 0.8%, respectively) [57]. As many as 10 of the 15 RAMs assess anticoagulant-associated bleeding risk for patients under VKA. The Skowrońska, Nieto, Chopard, and RIETE RAMs indistinctly include any treatment for VTE, from thrombolysis to cava filter, and all anticoagulant drugs, such as UFH, LMWH, DOAC, and VKA. The Kuijer and the Nieuwenhuis RAMs were developed using patients treated with UFH/LMWH. Five of the fifteen RAMs were internally validated on a population treated with DOAC: the VTE-BLEED RAM (dabigatran), the HOKUSAI RAM (edoxaban), the EINSTEIN RAM (rivaroxaban), the CAT-BLEED (edoxaban) and the Alonso RAM (apixaban and rivaroxaban). In recent years, clinical practice in VTE treatment has shifted towards a more common use of DOACs in cancer patients as a first choice or as an alternative to dalteparin, and VKAs are currently the anticoagulant least prescribed. Clinicians and RAMs should consider the difference in bleeding risk between VKA and DOAC when applying a RAM based on a population of VKA patients. To address this consideration, Alonso et al. proposed to add points in their RAM depending on the type of anticoagulant of the patients (DOAC or not DOAC), and the CAT-BLEED proposed to add points in its RAM if a patient is treated with edoxaban and has gastro intestinal cancer.

#### 3.2.3. What Is the Anticoagulant-Associated Bleeding Risk?

The principal outcome of the studies developing the discussed RAMs is the occurrence of bleeding events in CAT patients on anticoagulants, but what is really evaluated and what is worth evaluating are not always clearly defined. Bleeding events are by far the most frequent complication of anticoagulant therapy (25.8%, 95% CI: 24.8–26.8%) [58]. The bleeding events are usually categorized into major bleeding, clinically relevant non-major bleeding, and minor bleeding. Before 2005, a variety of definitions of major bleeding were used in published clinical studies, and this diversity further complicates data comparison across trials and in performing meta-analyses.

Of the 15 RAMs found in the literature, two were developed before 2005 (the Kuijer and Nieuwenhuis RAMs) and therefore did not implement the instituted ISTH definition. The RAM proposed by Nieto et al. [40] has a unique focus on fatal bleeding. The Alonso RAM, even though it is one of the most recently published studies, defined major bleedings using the International Classification of Diseases to categorize events, such as death, following a bleeding or symptomatic bleeding in a critical body area or organ. However, the criteria on blood supply cannot be taken into account with this method since a “blood transfusion” is not a disease (Table 4).

The occurrence of anticoagulant-associated bleeding has relevant prognostic and management implications. Major bleedings can be life-threatening and are generally associated with treatment discontinuation, which in turn contributes to adverse outcomes by leaving the patient exposed to an increased risk of thromboembolic recurrence. Non-major bleedings are more frequent than major bleedings and even if they do not directly lead to life-threatening situations, they are linked to a significant decrease in quality of life. If we focus on the rate of anticoagulant-associated bleedings in the populations used for developing the different RAMs, two features stand out. Firstly, the rate of major bleedings is lower than what would be expected based on the literature. This implies that the population employed had a lower risk of bleeding, which might explain the lower discriminatory potential of people at risk of major bleeding (2–5%) (Table 4) [59]. Secondly, most of the RAMs do not assess clinical non-relevant major bleedings (CNRMB), and therefore they do not allow for drawing conclusions regarding this type of bleeding. Surprisingly, only 5 of the 15 RAMs evaluated the risk of non-major bleeding. Nieuwenhuis et al. and Kuijer et al. introduced a RAM applicable to all bleeding events, with a distinction for major bleeding without specifying whether the non-major bleedings in their studies were clinically relevant or not. In 2015, ISTH introduced criteria to define CRNMBs for VTE studies [25]. The VTE-BLEED RAM was the first RAM published following these criteria. The Martinez RAM used a modified definition of CRNMB, consisting of bleeding events that resulted in hospitalization instead of all the CRNMBs as defined by ISTH. Skowrońska et al. presented the most recent RAM with a study of CRNMBs. It is important to clearly define whether the goal of the RAM is to assess the impact on quality of life or to assess life-threatening risk and to propose adequate care, pursuit, discontinuation, or reduction of the anticoagulant.

Finally, as mentioned previously, bleeding risk of patients treated with VKA or DOAC is different (bowel bleeding risk versus cerebral bleeding risk…), and differences might also exist among DOACs [57]. Therefore, anticoagulant-associated bleeding risk must be evaluated according to the type of anticoagulant.

#### 3.2.4. When Should Anticoagulant-Associated Bleeding Risk Be Assessed in CAT Patients?

The moment when the fatal anticoagulant-associated bleeding risk exceeds the risk of fatal PE (because later this is decreased) might be the best targeted time for anticoagulant-associated bleeding risk assessment for anticoagulant treatment indicated for 3 months or 6 months. Most bleeding events occur within the first few months of anticoagulant treatment, while the risk of VTE recurrence is also increased in parallel [60]. Factors influencing the risk of bleeding in the initial period of anticoagulation therapy may not be relevant for the prediction of anticoagulant-associated bleeding during extended treatment for the secondary prevention of recurrent VTE. Most of the RAMs evaluated their performance at the initiation of the treatment. The EINSTEIN, VTE Bleed, and Alonso RAMs evaluated the risk of bleeding 3 to 4 weeks after the diagnosis of VTE because after one month of therapy the risk of recurrent thrombosis drops dramatically while the risk of anticoagulant-associated bleeding remains stable (Table 4) [9]. Nevertheless, no international recommendation supports a treatment for VTE of less than 3 months, so it might be too early to assess the risk of bleeding at that time. A risk prediction rule for anticoagulant-associated bleeding in the setting of the prevention of the recurrence of VTE should ideally be targeted at patients who have already completed the initial length of treatment. Therefore, predictors of bleeding should preferably be measured at 3 months or 6 months rather than at anticoagulation initiation or 3 to 4 weeks after the VTE diagnosis. Moreover, only the Seiler RAM, the EINSTEIN RAM, and the HOKUSAI RAM were derived from a population with a follow-up of 3 months to 6 months, making the anticoagulant-associated bleeding risk assessment by most RAMs even less reliable for patients with anticoagulant treatment that is continued after 3 months or 6 months.

## 4. Discussion

The management of patients with CAT is challenging due to a higher risk of both recurrent VTE and bleeding events compared with non-cancer patients with VTE. By focusing on methodology and its applicability to CAT patients, the current review illustrates the limitations of the available risk assessment models for anticoagulant-associated bleeding in CAT.

Three systematic reviews on prediction models in VTE for anticoagulant-associated bleeding risk have been published. Refs. [18,61,62] Unlike these reviews, we specifically evaluated the risk of bleeding in CAT patients.

None of the presented RAMs developed in the general population were externally validated in a population of CAT patients. CAT BLEED, the only RAM derived from a population of CAT patients, seems promising as an anticoagulant-associated RAM for CAT patients but has also not been externally validated. Other issues that we encountered in some of the available RAMs were: the lack of standardization in the definitions of the types of bleeding; the differences in the time points that were chosen to assess the anticoagulant-associated bleeding; the large variety in types of anticoagulation agents used (VKA versus DOACs); the characteristics of the patients (cancer type, age, presence of renal failure or terminal illness) in clinical practice; and a large overlap in the predictors of VTE recurrence (Table 6)

The major concern lies in the timing of the risk assessment. We know from epidemiological studies performed in all VTE patients eligible for long-term treatment (cancer and cancer-free patients) that the case fatality rate for recurrent VTE seems to decrease during the initial 6 months of anticoagulant therapy from 11.3% (CI: 8.0% to 15.2%) to 3.6% (CI: 1.9% to 5.7%) and remain at this level after another 6 months. In contrast, the case fatality rate of a major bleeding event seems to remain constant over time (about 11.0%) [60]. Therefore, estimating the bleeding risk at treatment initiation is too early for a patient that will remain on anticoagulation therapy for 3 months or 6 months. An early assessment could lead to differences in the characteristics between patients who might die from PE during the onset of the anticoagulation therapy and those who survive beyond the 3 months or 6 months of treatment and might also account for some of the apparent differences in the case fatality rates. The best strategy might be to design different RAMs for before and after the 3 months or 6 month thresholds.

CAT patients are at high risk of VTE. Medical care is now changing from treating all patients at high risk of VTE to benefit-based treatment, which means that only those patients who will have a greater benefit from more from treatment than the risk of major bleeding will be treated [63]. Since the risk of associated anticoagulant bleeding is also high in the CAT population, categorizing the patients that will benefit from the correct anticoagulation therapy in terms of duration or intensity is important. Anticoagulant indication and prescription may improve when treatment duration is decided based on an individualized evaluation between the absolute recurrence risk reduction and the absolute increase in bleeding risk, which should be the aim of future development studies.

## 5. Conclusions

Since bleeding complications are associated with significant morbidity and a decrease in quality of life in patients with cancer, in order to minimize these complications it is important to weigh the risks and benefits of treating patients with anticoagulation therapy [10,11]. Optimization of the assessment of bleeding risk in this specific population needs to be undertaken. Currently available prediction models for bleeding during anticoagulant treatment after VTE have important methodological limitations, insufficient predictive accuracy, or lack independent external validation when applied to CAT patients in decision making. This review gives an overview of the currently available RAMs for anticoagulant-associated bleeding in CAT patients. Another approach should be taken to develop a RAM with good performance when it comes to external validation.

## Figures and Tables

**Figure 1 cancers-14-01937-f001:**
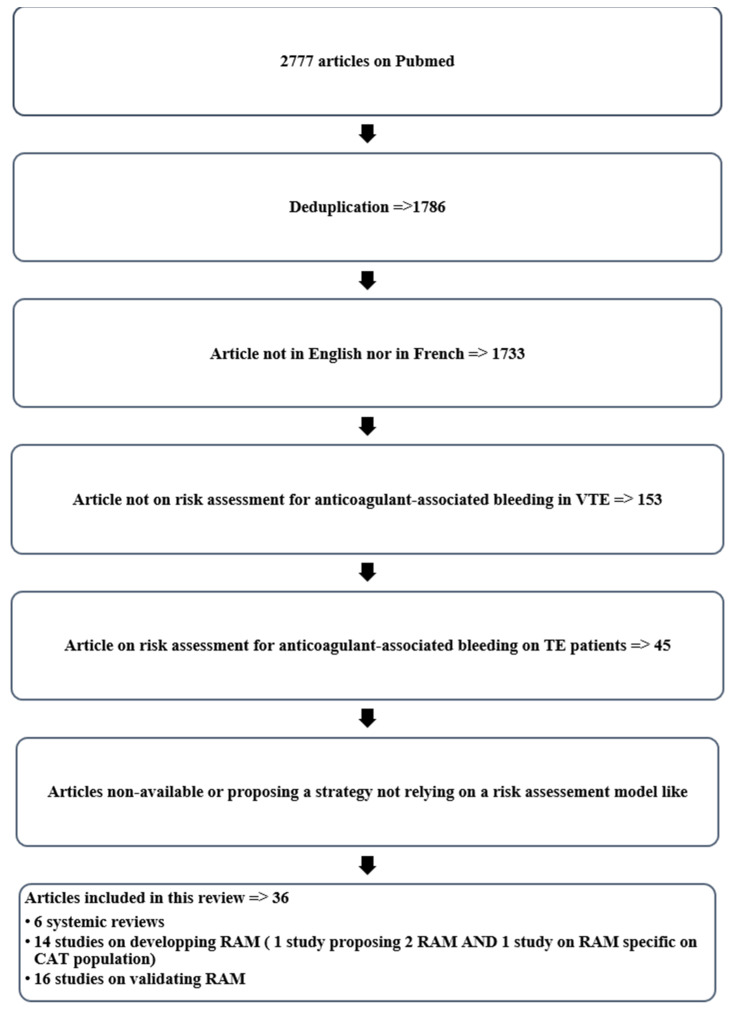
Flowchart of the study selection.

**Table 1 cancers-14-01937-t001:** Pilot question form.

Criteria	Question
Bleeding risk assessment	Risk of bleeding during anticoagulation
Participants	All adult patients with VTE
Risk factors	Patients’ demographics, cancers, comorbidities, concomitant treatments, physiological variables, laboratory measurements, genetics, and history of bleeding before the index event
Outcome to be predicted	Bleeding, ISTH major bleeding, fatal bleeding, and clinically relevant non-major bleeding

**Table 2 cancers-14-01937-t002:** Electronic search strategy for PubMed research through 20 September 2021 with no date restriction.

Venous Thromboembolism	Prediction	Bleeding
Venous thromboembolism [MeSH] OR	Clinical prediction rule [MeSH] OR	Bleeding [MeSH]
Pulmonary embolism [MeSH] OR	Risk management model [MeSH] OR	OR
Venous thrombosis [MeSH] OR	Prognostic score [MeSH] OR	Hemorrhage [MeSH] OR hemorrhage [MeSH]
Deep vein thrombosis [MeSH] OR	Prediction score [MeSH]	-

**Table 3 cancers-14-01937-t003:** Prediction factors for anticoagulant-associated bleeding RAMs.

Characteristics	ACCP [16]	EINSTEIN [37]	HOKUSAI [38]	Kuijer [33]	Martinez [32]	Nieuwenhuis [28]	RIETE [27]	VTE-BLEED [29]	Seiler [34]	CAT-BLEED [33]	Alonso [39]	Nieto [40]	Chopard [31]
**Demographic characteristics**													
Age	X	X		X	X		X	X		X	X	X	
Sex (Female, F or Male, M)		X	(F)	(F)	(M)			(M)			(F)		
BMI					X	X							
Race		X											
**Bleeding risk factors**													
Alcohol abuse	X										X	X	
History of bleeding	X				X	X	X	X	X		X	X	X
Kidney and/or liver failure	X				X		X	X		X	X		
Diabetes mellitus	X										X		
Uncontrolled hypertension. (+/− Male)		X	X					X					
Recent surgical procedure	X					X							
Antiplatelet therapy and NSAIDs	X		X						X		X		
Poor anticoagulant control	X								X				
Frequent falls, previous stroke, dementia	X				X								
Recent trauma					X	X							
**Cancer history**													
(active) Cancer or metastatic cancer	X			X	X		X	X	X	X	X	X	
Genitourinary cancer										X			
Gastrointestinal cancer and Edoxaban treatment										X			
Anticancer therapy with gastrointestinal toxicity										X			
**Index events**													
Pulmonary embolism as index event					X		X						
Distal DVT												X	
**Other comorbidities**													
Comorbidity + decrease in functional capacity/immobility	X								X			X	
Cardiovascular disease (stroke/coronaropathy/peripheral arterial disease)	X	X			X						X		
Syncope													X
Tobacco and COPD					X						X		
**Biological parameters**													
Anemia/Hemoglobin	X	X	X				X	X	X		X		X
INR/abnormal prothrombin time	X								X			X	
Thrombopenia	X							X			X	X	
D-dimer							**X ***	**X ***					
**Drugs**													
Rivaroxaban											X		
Apixaban											X		
VKA											X		

X * The 2 Skowrońska [30] RAMs are modified version of the VTE BLEED RAM and the RIETE RAM in which D-dimer assessment were added in the RAMS.

**Table 4 cancers-14-01937-t004:** Risk assessment model developed to assess bleeding risk.

Reference	Type of Sources	Follow-Up (Months)	Time of Inclusion	Anticoagulant Type	Number of Patients	Number of Patients with Active Cancer (%)	Bleeding Outcome	Number of Bleedings	OR/HR/RR/β-Coefficient Cancer	Limiting Exclusion Criteria
NIEUWENHUIS Nieuwenhuis et al., 1991 [28]	Randomized, controlled trial	<1	Baseline	LMWH	96	64 (32.9%)	Major bleeding (death, interruption of treatment, transfusion, a decrease of >2.42 g/dL) and minor bleeding (= non major bleeding)	23 Major bleedings	RR: 0.9	-
				UFH	98					
KUIJER Kuijer et al., 1999 [33]	Randomized controlled trial	3	Baseline	LMWH	510	119 (23%)	All bleeding episodes during anticoagulation, Major bleeding (critical site, interruption of treatment, transfusion, a decrease of >2.42 g/dL)	16 Major bleedings (46 total bleeding)	OR: 2.2 (/)	-
				UFH	511	113 (22%)		12 Major bleedings (47 total bleeding)		
RIETE Ruiz-Giménez et al., 2008 [27]	Prospective cohort	3	Baseline	LMWH/ UFH or VKA or Cava filter	13,057 (derivation sample) and 6572 (validation sample)	2 756 (21.1% of the derivation sample) and 1321 (20 % of the validation sample)	Major bleeding (ISTH) during anticoagulation	111 Major bleedings and 337 Non major bleeding	OR: 2.1 (1.7–2.6)	-
NIETO Nieto et al., 2010 [40]	Prospective cohort	3	Baseline	Thrombolytic/LMWH/ UFH or VKA or Cava filter	24395	5063 (20.8%)	Fatal bleeding	135 Fatal bleeding	OR: 2.87 (2.04–4.03)	Patients currently participating in a therapeutic clinical trial with a blinded therapy
EINSTEIN Di Nisio et al., 2016 [37]	Randomized controlled trial	3 to 12	I within the 3 weeks, and after the first 3 weeks, overall study	RIVAROXABAN	4130	232 (5.6%)	Major bleeding (ISTH) during anticoagulation	40 Major bleedings	HR: 3.47 (1.79–6.7) in the 3 first weeks and HR: 2.49 (1.54–4.03) for the entire study	Creatinine clearance < 30 mL/minute/Clinically significant liver disease/Active bleeding or a high risk of bleeding contraindicating anticoagulant treatment/Uncontrolled high blood pressure/ Life-expectancy of <3 months
				LMWH/VKA	4116	196 (4.8%)		72 Major bleedings		
VTE-BLEED Klok et al., 2016 [29]	Randomized, controlled trial	6	1 month, overall study	DABIGATRAN	2553	114 (2.2%)	Major bleeding (ISTH) and CRNMB (ISTH) during anticoagulation	37 Major bleedings and 101 CRNMB	OR: 4.18 (2.50–7.02)	High risk of bleeding/Liver disease/Creatinine clearance < 30 mL per minute/Life expectancy of less than 6 months/Requirement for long-term antiplatelet therapy > 100 mg of aspirin
				VKA	2554	-		51 Major bleedings and 167 CRNMB	-	
ACCP Kearon et al., 2016 [16]	Meta-analysis of 9 studies	6	-	LMWH and VKA	3637	-	Major bleeding (ISTH) during anticoagulation	-	RR: 0.96 (0.65–1.42)	-
SEILER Seiler et al., 2017 [34]	Prospective cohort	36	Baseline	VKA	1003	71 (<1%)	Major bleeding (ISTH) during anticoagulation	66 Major bleedings (743 bleedings)	β-coefficient: 0.56 (−0.18–1.3)	Terminal illness/Catheter-related thrombosis
HOKUSAI Di Nisio et al., 2017 [38]	Randomized, controlled trial	3 to 12	Baseline	LMWH/ EDOXABAN	4118	109 (4.59%)	Major bleeding (ISTH) during anticoagulation	56 Major bleedings	OR: 3.86 (1.50–9.92)	Cancer for which long-term treatment with LMWH was anticipated/ Aspirin at a dose > 100 mg daily or dual antiplatelet therapy/ Creatinine clearance < 30 mL/min
				LMWH/VKA	4122	99 (3.03%)		66 Major bleedings	OR: 2.17 (0.67–7.06)	
				LMWH	522	511 (97.5%)	-	-	-	-
Skowrońska et al., 2019 [30]	Prospective cohort	0.5	Baseline	Thrombolysis, UHF, LMWH, FONDAPARINUX,	310	57 (18.3%)	Major bleeding (ISTH) and CRNMB (ISTH) during anticoagulation that occurred during the hospital stay	18 Major bleedings and 17 CRNMB	-	-
				RIVAROXABAN, VKA						
	Prospective cohort		Baseline	Combination therapy (VKA + LMWH)						
MARTINEZ Martinez et al., 2019 [32]	Prospective cohort	3	Baseline	VKA	10,010	746 (7.45%)	Major bleeding and CRNMB resulting in hospitalization (CRNMB-H)	344 Major bleedings and 3 112 CRNMB-H	Early post-VTE active cancer sHR: 7.92 (4.33–14.49) Persisting active cancer sHR: 1.69 (0.99–2.88)	≥2 VKA prescriptions before the initial VTE diagnosis
Chopard et al., 2021 [31]	Prospective cohort	1	Baseline	UFH, LMWH, DOAC, cava filters, thrombolysis	2754	507 (18%)	Major bleeding (ISTH) during anticoagulation	82 Major bleedings	-	-
Alonso et al., 2021 [39]	Database from 2011 to 2017	6	Initially 4 weeks, after VTE	VKA	116,319	18 (<1%)	Hospitalization for intracranial hemorrhage, gastrointestinal bleeding, or other major bleeding as defined by the International Classification of Diseases, (9th and 10th)	2294 bleedings	HR: 1.43 (1.30–1.47)	Patient using dabigatran 1141
				RIVAROXABAN	37,214	16 (<1%)				
				APIXABAN	11,901	17 (<1%)				
CAT-BLEED Winter et al., 2021 [33,41]	Randomized, controlled trial	6	Baseline	EDOXABAN	524	513 (98.3%)	Major bleeding (ISTH) during anticoagulation	39 Major bleedings and 110 CRNMB	Genitourinary cancer sHR: 2.48 (1.14–5.38)	Active bleeding/ Aspirin at a dose > 100 mg daily or dual antiplatelet therapy/ Creatinine clearance < 30 mL/min/ Clinically significant liver disease/ Uncontrolled high blood pressure/ ECOG 3–4/Life expectancy < 3 month/ Platelet count < 50,000
				LMWH	522	511 (97.5%)			Gastrointestinal cancer edoxaban treatment sHR: 2.20 (1.07–4.53) Regionally advanced or metastatic cancer sHR: 1.21 (0.82–1.80)	

**Table 5 cancers-14-01937-t005:** Validation studies for risk assessment model.

Reference	Bleeding Model	AF Model	Type of Sources	Follow Up (Months)	Anticoagulant Type	Number of Patients	Bleeding Outcome	Number of Major Bleeding	Number of Patient with Active Cancer (%)	Bleeding Outcome in Cancer Patients	OR/HR/RR/β-Coefficient Cancer	Limiting Exclusion Criteria	Conclusion of the Author on the Validation in Clinical Practice
Scherz et al., 2013 [42]	ACCP, Kuijer, RIETE	OBRI	Prospective cohort, multicenter	3	Thrombolysis, UHF, LMWH, FONDAPARINUX	663	Major bleeding (ISTH) during anticoagulation	28	98 (14.6%)	9	-	Patients <65 y.o.	-
					VKA								
Nieto et al., 2013 [43]	Nieto	-	Prospective cohort, multicenter (RIETE)	-	Thrombolysis/LMWH/ UFH or VKA or Cava filter	15,206	Fatal bleeding	52	3468 (22.8%)	29	-	Patients currently participating in a therapeutic clinical trial with a blinded therapy	better for predicting gastrointestinal than intracranial fatal bleeding
Poli et al., 2013 [44]	ACCP 2012, RIETE	ATRIA, HAS-BLED, HEMORR2HAGES, OBRI,	Prospective cohort (EPICA); 27 hospitals in Italy	24	VKA	887	Major bleeding (ISTH) during anticoagulation	47	110 (10.1%)	11	1.1 (0.6–2.3)	Judged too frail	No
Riva et al., 2014 [45]	ACCP 2012, Kuijer, RIETE	ATRIA, HAS-BLED, HEMORR2HAGES, Shireman	Retrospective cohort; anticoagulation clinics of 5 hospitals in Italy	12	VKA	681	Major bleeding (ISTH) and CRNMB (ISTH) during anticoagulation	50	78 (11.4%)	/	-	-	No
Piovella et al., 2014 [46]	RIETE, KUIJER	mOBRI	Prospective cohort, multicenter (RIETE)	3	Thrombolysis, UHF, LMWH,	8717	Major bleeding = clinically overt with a need for transfusion of at least two units of red blood cells/retroperitoneal or intracranial/ permanent discontinuation of treatment/ fatal	82	1807 (20.7%)	22	-	-	Slightly better performance of the RIETE
		OBRI			RIVAROXABAN, VKA								
Kline et al., 2016 [47]	RIETE, KUIJER	mOBRI OBRI	Pooled data of EINSTEIN PE and EINSTEIN DVT	3 to 12	RIVAROXABAN	4130	Major bleeding (ISTH) during anticoagulation	40	232 (5.6%)	-	-	-	Good performance for RIETE
Klok et al., 2017 [48]	VTE-BLEED	-	RCT (HOKUSAI VTE) international study	3 to 12	VKA	3903	Major bleeding (ISTH) during chronic, stable anticoagulation (>30 days)	40	181 (31%)	6	-	-	Yes
Palareti et al., 2018 [49]	ACCP 2016	-	Prospective cohort (START2) in multiple hospitals in Italy	>12	VKA DOAC (subtype not specified)	2263	Major bleeding (ISTH) and CRNMB (ISTH) during anticoagulation	48	175 (23.4%)	4	HR = 1.0 (0.4–3.0)	-	No
Rief et al., 2018 [50]	VTE-BLEED	HAS-BLED	Prospective cohort study, 1 hospital in Austria	12	LMWH, VKA, APIXABAN, RIVAROXABAN, EDOXABAN,	111	Major bleeding (ISTH) during anticoagulation	4	12 (11%)	-	-	-	Did not discuss validity of the VTE bleed
Zhang et al., 2018 [51]	ACCP, Kuijer, RIETE, NIEUWENHUIS	-	Prospective cohort	3	VKA, LMWH	563	Major bleeding (ISTH) and CRNMB (ISTH) during anticoagulation	16	70 (12.4%)	-	-	-	Good performance of the ACCP
Klok et al., 2018 [52]	VTE-BLEED	-	RCT (Xalia); multiple hospitals in 12 countries	>12	LMWH RIVAROXABAN	4457	Major bleeding (ISTH) during anticoagulation	39	500 (11%)	-	HR = 1.0 (0.61–1.7)	-	Yes
Vedovati et al., 2019 [53]	Kuijer, RIETE, VTE-BLEED,	HAS-BLED, ATRIA	Prospective cohort	>12	APIXABAN, RIVAROXABAN, EDOXABAN, DABIGATRAN	1034	Major bleeding (ISTH definition) during anticoagulation	26	164 (15.9%)	5	HR = 1.930 (0.721–5.170)	-	No
Skowrońska et al., 2019 [30]	VTE-BLEED, RIETE	HEMORR2HAGES, HAS-BLED	PE-aWARE registry	0.5	Thrombolysis, UHF, LMWH, FONDAPARINUX,	310	Major bleeding (ISTH) and CRNMB (ISTH) during anticoagulation that occurred during the hospital stay	17	56 (18.1%)	11	-	-	Good performance at identifying Acute PE patients at risk of in-hospital bleeding complication of the VTE bleed
					RIVAROXABAN, VKA								
					Combination therapy (VKA + LMWH)								
Keller et al., 2021 [54]	KUIJER	-	Nationwide German registry	-	DOAC, VKA	1,204,895	Hospitalization for intracranial hemorrhage, gastrointestinal bleeding, or other major bleeding as defined by the International Classification of Diseases	-	25885 (2.1%)	-	-	-	Good performance at predicting in hospital major bleeding
Mathonier et al., 2021 [55]	VTE-BLEED, RIETE	ORBIT, HEMORR2HAGES, ATRIA, HAS-BLED	BFC-FRANCE registry	0.25	UFH, LMWH, FONDAPARINUX, VKA and DOACs	2754	Major bleeding (ISTH) that occurred during the hospital stay	82	507 (18.4%)	17	OR= 4.7 (3.2–6.8)	-	No
Frei et al., 2021 [56]	VTE-BLEED, Seiler, Kuijer, RIETE, ACCP,	OBRI, HEMORR2HAGES, HAS-BLED, ATRIA	Prospective, multicenter SWIss venous Thromboembolism COhort study 65+ (SWITCO 65+)	36	VKA	743	Major bleeding (ISTH) and CRNMB (ISTH) during anticoagulation	45	10 (1.3%)	16	-	Terminal illness, catheter-related thrombosis, age under 65	No
De Winter et al., 2021 [36]	VTE-BLEED, RIETE, Martinez, Kuijer, HOKUSAI, ACCP	HAS-BLED	HOKUSAI VTE cancer post hoc analysis	>12	EDOXABAN, LMWH	1046	Major bleeding (ISTH) and CRNMB (ISTH) during anticoagulation that occurred during the hospital stay	39	1024 (97.8%)	39	-	-	No good performance of the existing RAM in CAT population

**Table 6 cancers-14-01937-t006:** Prediction factors of recurrent VTE risk and anticoagulant-associated bleeding risk [18].

Ambivalent Prediction Factors of Anticoagulant-Associated Bleeding and Recurrent VTE
Age
Sex (Female or Male)
BMI
PE as the index VTE
History of cardiovascular disease (stroke/coronaropathy/peripheral arterial disease)
Cancer site
Cancer stage
Chemotherapy

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
