# Peer review of "In Search of the Appropriate Anticoagulant-Associated Bleeding Risk Assessment Model for Cancer-Associated Thrombosis Patients"

_cancers, 2022, doi:10.3390/cancers14081937_

Round 1

Reviewer 1 Report

Methods:

The main problem of the article is the application of the PRISMA checklist which primarily focuses on the reporting of reviews evaluating the effects of interventions, although it can also be used as a basis for reporting systematic reviews with objectives other than evaluating interventions (e.g. evaluating aetiology, prevalence, diagnosis or prognosis).

The PRISMA checklist ( see page  18) requires evaluation of Risk of bias across which however was not  conducted, as prognostic scores are involved and not interventions. As the topic involves prognostic models, result the PROBAST tool ( Prediction model study risk of bias assessment tool - Ann Intern Med. 2019;170:W1-W33. doi:10.7326/M18-1377;  Moons KGM) seems more appropriate to assess the quality of  included studies.

In addition after the systematic literature search, this seems to be  a narrative review as no validated score for bleeding risk assessment is available for cancer associated thrombosis.

Author Response

Dear reviewer,

First of all, we would like to thank you for the careful review of our work entitled: “In Search for the Appropriate Anticoagulant Associated Bleeding Risk Assessment Model for Cancer Associated Thrombosis Patients”.

  • Comment 1 from reviewer 1 (C1R1)

“The main problem of the article is the application of the PRISMA checklist which primarily focuses on the reporting of reviews evaluating the effects of interventions, although it can also be used as a basis for reporting systematic reviews with objectives other than evaluating interventions (e.g. evaluating etiology, prevalence, diagnosis or prognosis).The PRISMA checklist ( see page 18) requires evaluation of Risk of bias across which however was not conducted, as prognostic scores are involved and not interventions. As the topic involves prognostic models, result the PROBAST tool (Prediction model study risk of bias assessment tool - Ann Intern Med. 2019;170: W1-W33. doi:10.7326/M18-1377; Moons KGM) seems more appropriate to assess the quality of included studies.”

  • Response to C1R1

As you have suggested, we have incorporated the evaluation of the risks of bias of our prognostic models with the PROBAST (Appendix A)

Moreover, in the results section, we have detailed the risk of bias of these different scores with the following sentences at the end of the results section page 4:

“Included model derivation studies were generally at low risk of bias for 'predictors' and 'outcome'. However, for the population, as these RAMs derived mainly from randomized trials, the percentage of CAT patients was often low and their specific risk was not analyzed. With regard to the "statistical analysis" domain, almost all the derivation studies included were judged to be at high risk of bias due to poor management of the parameters in continuous values ​​and the low event rates. These commentaries are transposable to RAM validation studies.”

  • Comment 2 from reviewer 1 (C2R1)

“In addition, after the systematic literature search, this seems to be a narrative review as no validated score for bleeding risk assessment is available for cancer associated thrombosis.”

  • Response to C2R1

Finally, we agree with Reviewer 1 that our study demonstrates the absence of a strong validated RAM for the prediction of anticoagulant associated bleeding risk, and call for the development of better tools. The discussion and conclusion support this conclusion.

Reviewer 2 Report

In this manuscript, the authors nicely reviewed many publications to investigate the anticoagulant associated bleeding RAM, and their applicability in the CAT patients.

Overall, this is a very nice review with large amount of literature search. The authors point out some limitations of current RAM for accessing the anticoagulant associated bleeding risk in CAR patients.

1. Can the authors discuss more what are the limitations of current RAM for CAT patient in the Results part? And why they are not appropriate for the CAT patients?

2. The authors proposed that adding “WHOM”, “HOW”, “WHAT”, “WHEN” questions may be more appropriate to access anticoagulant-associated bleeding risk in CAT patients. Can the authors try to add these questions into some current RAM model, make a table, and compare the new model with the old model?

Author Response

  • Response to reviewer 2

Dear reviewer,

We wanted to thank you first of all for your attentive reading and your very positive feedback for our review:” In Search for the Appropriate Anticoagulant Associated Bleeding Risk Assessment Model for Cancer Associated Thrombosis Patients”.

  • Comment 1 from reviewer 2 (C1R2)

“Can the authors discuss more what are the limitations of current RAM for CAT patient in the Results part? And why they are not appropriate for the CAT patients?”

  • Response to C1R2

To address more about the limitation of the different RAM in the results section, we have detailed the risk of bias of these different prognostic RAM with the following sentences at the end of the results section page 4:

“Included model derivation studies were generally at low risk of bias for 'predictors' and 'outcome'. However, for the population, as these RAMs derived mainly from randomized trials, the population of cancer patients was most often low and their own risk was not analyzed. With regard to the "statistical analysis" domain, almost all the derivation studies included were judged to be at high risk of bias due to poor management of the parameters in continuous values ​​or due to low event rates. These commentaries are almost transposable to RAM validation studies.”

  • Comment 2 from reviewer 2 (C2R2)
  1. The authors proposed that adding “WHOM”, “HOW”, “WHAT”, “WHEN” questions may be more appropriate to access anticoagulant-associated bleeding risk in CAT patients. Can the authors try to add these questions into some current RAM model, make a table, and compare the new model with the old model?
  • Response to C2R2

You will find attached a table summarizing the ability of each score to answer the questions which is also proposed as a visual abstract to the journal.

in WHOM = CAT patients

HOW = on an adpated anticoagulant regimen 

WHAT = Bleeding as defined by ISTH

WHEN = after 3 months

Yes

NIEUWENHUIS Nieuwenhuis 1991[28]

SEILER Seiler 2017[34]

RIETE Ruiz-Giménez 2008[27]

KUIJER Kuijer 1999[33]

HOKUSAI Di Nisio 2017[46]

EINSTEIN Di Nisio 2016[45]

RIETE Ruiz-Giménez 2008[27]

Chopard 2021[31]

VTE-BLEED Klok 2016[29]

Skowrońska 2019[30]

Alonso 2021[47]

ACCP Kearon 2016[16]

MARTINEZ Martinez 2019[32]

CAT-BLEED Winter 2021[33,37]

SEILER Seiler 2017[34]

Chopard 2021[31]

HOKUSAI Di Nisio 2017[46]

Alonso 2021[47]

Skowrońska 2019[30]

Chopard 2021[31]

CAT-BLEED Winter 2021[33,37]

More or Less

NIETO Nieto 2010[48]

EINSTEIN Di Nisio 2016[45]

NIEUWENHUIS Nieuwenhuis 1991[28]

SEILER Seiler 2017[34]

KUIJER Kuijer 1999[33]

CAT-BLEED Winter 2021[33,37]

NIETO Nieto 2010[48]

MARTINEZ Martinez 2019[32]

Alonso 2021[47]

No

EINSTEIN Di Nisio 2016[45]

NIEUWENHUIS Nieuwenhuis 1991[28]

All of them

VTE-BLEED Klok 2016[29]

KUIJER Kuijer 1999[33]

HOKUSAI Di Nisio 2017[46]

RIETE Ruiz-Giménez 2008[27]

VTE-BLEED Klok 2016[29]

ACCP Kearon 2016[16]

NIETO Nieto 2010[48]

Skowrońska 2019[30]

MARTINEZ Martinez 2019[32]

N.A

ACCP Kearon 2016[16]